# Scaling Laws: A Model-Based Optimization Perspective

## Abstract

Scaling laws have become indispensable for guiding the pre-training of large language models, enabling optimal decision-making—such as determining the scale of model and data—under a fixed compute budget. Standard practice involves fitting parametric functions (predominantly power laws) from small-scale experiments, which allows researchers to extrapolate trends and predict compute-optimal configurations at larger scales. This neural scaling paradigm is fundamentally a specialized instantiation of *Model-Based Optimization* (MBO): constructing a surrogate model (the scaling law) from experimental data to predict validation metrics, and subsequently optimizing pre-training configurations as design variables against this surrogate. Despite this equivalence, existing literature primarily focuses on neural scaling priors while neglecting the broader MBO perspective. In this position paper, we bridge this gap by formally mapping the neural scaling paradigm to the three stages of MBO: *design space*, *surrogate modeling*, and *guided optimization*, and distinguish the three unique characteristics— low-dimensional spaces, strong power-law priors, and strict compute constraints—that separate it from standard MBO problems. Furthermore, we systematically partition the design space into three subspaces: *model*, *data*, and *hyperparameters*. Crucially, we formalize their relationship as a bi-level optimization problem, wherein hyperparameters are optimized at the lower level to ensure convergence for specific model and data configurations. To demonstrate the practical utility of adopting MBO techniques, we focus on the surrogate modeling stage and provide an illustrative proof-of-concept by applying *autofocus*—an established MBO technique—to mitigate extrapolation-induced covariate shifts. Finally, we conclude by providing a principled roadmap for future research, highlighting uncertainty quantification and multi-objective optimization.

## 1 Introduction

The pre-training of modern large language models is governed by a fundamental tension between maximizing performance and managing prohibitive compute costs, which can run into the tens of millions of dollars for a single training run (Dubey et al., 2024; Team et al., 2023). In this high-stakes regime, scaling laws have become indispensable for guiding optimal decision-making, specifically, determining pre-training configurations, such as the scale of model and data, under strict compute budgets (Kaplan et al., 2020; Henighan et al., 2020; Hoffmann et al., 2022; Dubey et al., 2024). Standard practice involves fitting parametric functions, predominantly power laws (Hestness et al., 2017; Bahri et al., 2024), to validation metrics from small-scale experiments, and subsequently extrapolating these functions to predict compute-optimal configurations at larger scales. This empirical predictability allows researchers to forecast the behavior of billion-parameter models, effectively bypassing the prohibitive expense of full-scale trial and error.

This neural scaling paradigm is fundamentally a specialized instantiation of *Model-Based Optimization* (MBO) (Kim et al., 2026; Trabucco et al., 2022). MBO is a well-established framework in scientific and engineering domains for optimizing complex systems where obtaining ground-truth labels for designs is resource-intensive. The standard workflow involves constructing a surrogate model from an offline dataset of design-label pairs—for example, mapping robot morphology (design) to locomotion speed (label)—and subsequently optimizing the design variables against this surrogate to maximize the predicted label. The equivalence here is distinct: by mapping pre-training configurations (such as model and data scale) to *design*

*variables*, validation metrics to the *label*, and the parametric power-law fit to the *surrogate model*, the neural scaling paradigm is formally isomorphic to an MBO problem.

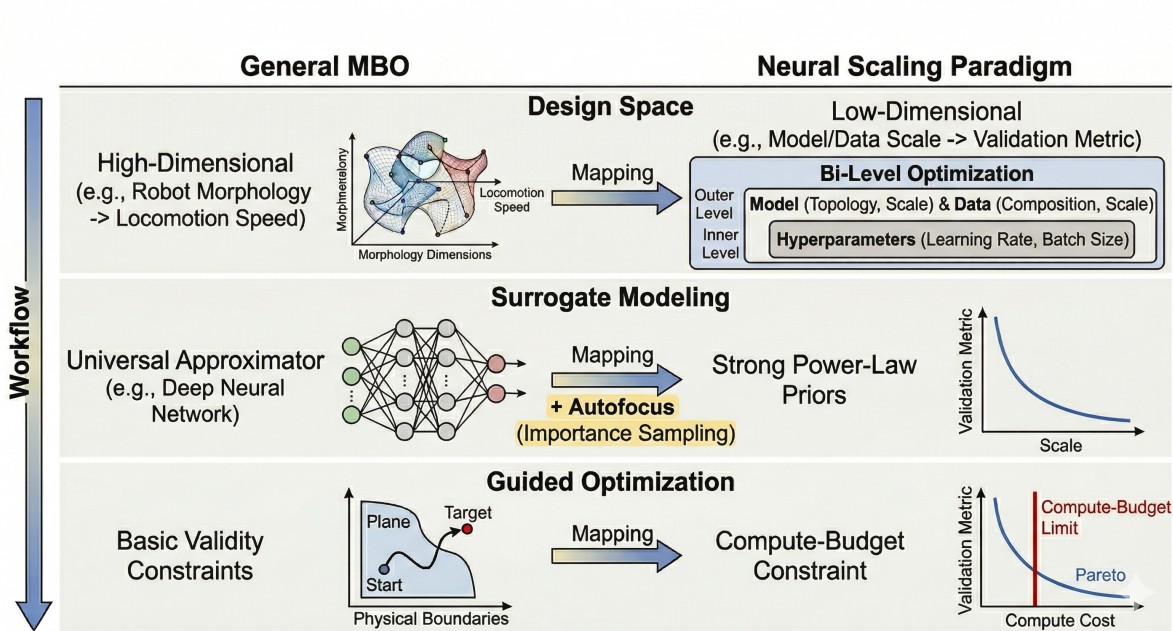

Figure 1: Overview Structure.

Despite this structural equivalence, existing literature remains fixated on domain-specific heuristics rather than the broader optimization framework. Recent research has concentrated on refining the design space—ranging from debates over the exclusion of embedding parameters (Kaplan et al., 2020; Hoffmann et al., 2022) to quantifying the impact of data repetition and quality (Muennighoff et al., 2023), and identifying optimal learning rate schedules (Hu et al., 2024). While these empirical contributions are essential for constructing accurate surrogate models, they often neglect the methodological rigors of the general MBO view. Crucially, established MBO techniques remain largely unexplored in this context; for instance, challenges like data mixture optimization could be addressed more systematically through uncertainty quantification and multi-objective optimization to navigate non-convex performance trade-offs.

In Section 2, we formally bridge this gap by mapping the neural scaling paradigm to the three stages of MBO: *design space*, *surrogate modeling*, and *guided optimization*, as illustrated in Figure 1. We then delineate the three unique characteristics—low-dimensional spaces, strong power-law priors, and strict compute constraints—that separate it from standard MBO problems. In Section 3, we systematically partition the design space into three subspaces: *model*, *data*, and *hyperparameters*. Crucially, we formalize their relationship as a bi-level optimization problem, wherein hyperparameters (e.g., learning rate and batch size) are optimized at the lower level to ensure convergence for specific model and data configurations. Within the model subspace, we examine how topology (e.g., architecture and multimodal fusion) and scale (e.g., parameter count and precision) collectively shape the surrogate landscape. Concurrently, we analyze the data subspace through a symmetric lens, detailing the impact of data composition (e.g., tokenizer design and mixture ratios) and effective data scale (e.g., token count and repetition).

In Section 4, to demonstrate the practical utility of adopting MBO techniques, we provide an illustrative proof-of-concept focusing on the surrogate modeling stage. Specifically, we apply *autofocus* (Fannjiang & Listgarten, 2020)—an importance sampling technique—to reweight the training data during surrogate fitting. This explicitly mitigates extrapolation-induced covariate shifts, minimizing prediction errors in data-constrained extrapolation regimes. Finally, in Section 5, we outline a principled roadmap for future

research, highlighting uncertainty quantification for active, risk-aware compute allocation, and multi-objective optimization for navigating the non-convex Pareto frontiers of competing capabilities.

## 2 Neural Scaling Paradigm as Model-Based Optimization

### 2.1 The General MBO Framework

*Model-Based Optimization* (MBO) addresses the problem of finding an optimal design $\boldsymbol{x}$ to maximize a black-box objective function $\boldsymbol{y} = \boldsymbol{f}(\boldsymbol{x})$, particularly when querying $\boldsymbol{f}(\boldsymbol{x})$ is computationally expensive and access is limited to an offline dataset. The MBO workflow is structurally composed of three stages:

1. **Design Space:** We first establish the label space $\mathcal{Y} \subseteq \mathbb{R}^m$, encompassing the $m$-dimensional objective values to be optimized. Next, we identify the specific variables that influence these objectives, thereby defining the $n$-dimensional design space $\mathcal{X} \subseteq \mathbb{R}^n$. An offline dataset $\mathcal{D} = \{(\boldsymbol{x}_i, \boldsymbol{y}_i)\}_{i=1}^{|\mathcal{D}|}$ is then collected, where each entry pairs a specific design configuration with its ground-truth label.

2. **Surrogate Modeling:** A surrogate model $\hat{f}_{\boldsymbol{\theta}} : \mathcal{X} \rightarrow \mathcal{Y}$ is trained on $\mathcal{D}$ to approximate the true objective landscape. The primary goal is to minimize the empirical prediction error over the existing dataset, effectively providing a differentiable or easily queryable proxy for the expensive ground-truth function.

3. **Guided Optimization:** With the surrogate $\hat{f}_{\boldsymbol{\theta}}$ fixed, we search for the optimal design $\boldsymbol{x}^*$ that maximizes the predicted performance. In single-objective optimization scenarios ($m = 1$), this is formally defined as:

$$\boldsymbol{x}^* = \arg\max_{\boldsymbol{x} \in \mathcal{X}} \hat{f}_{\boldsymbol{\theta}}(\boldsymbol{x}) \tag{1}$$

**Example: Robot Design.** Consider the task of optimizing a robot's morphology to maximize locomotion speed ($m = 1$). Here, the scalar $y$ represents the speed measured in a physical simulator, such as MuJoCo (Trabucco et al., 2022). An MBO pipeline involves: (1) *Design Space*, where we determine the morphological variables influencing speed—for instance, defining thigh length and shin length to form the design space ($n = 2$)—and collect an offline dataset $\mathcal{D}$ pairing various robot configurations with their resulting speeds; (2) *Surrogate Modeling*, where a neural network $\hat{f}_{\boldsymbol{\theta}}$ is fitted to predict speed directly from the morphological design; and (3) *Guided Optimization*, where gradient ascent is performed on the design variables $\boldsymbol{x}$ with respect to the surrogate's output to discover a faster robot configuration.

### 2.2 Stage Mapping: Formulating Scaling as MBO

The neural scaling paradigm, which aims to optimize performance under a fixed compute budget, can be framed as a specialized instantiation of MBO. The workflow formally maps to the three core stages of the general MBO framework:

1. **Design Space:** We first identify the objective $\mathcal{Y}$ as a scalar validation metric ($m = 1$), which can be the autoregressive validation loss or a downstream benchmark score. The design space $\mathcal{X}$ comprises the pre-training configurations contributing to this metric, which we partition into three subspaces: model, data, and hyperparameters (detailed in Section 3). The offline dataset $\mathcal{D}$ consists of historical training runs, pairing specific configurations (the design variables) with their empirical metrics (the labels).

2. **Surrogate Modeling:** The surrogate model $\hat{f}_{\boldsymbol{\theta}}$ is constructed to approximate the validation metric based on the input design variables. Unlike general MBO, this surrogate leverages strong functional priors—most notably parametric power laws—to relate the validation metric to continuous scale variables.

3. **Guided Optimization:** With the surrogate $\hat{f}_{\boldsymbol{\theta}}$ fixed, we optimize the design variables against this surrogate to search for the compute-optimal configuration $\boldsymbol{x}^*$ that minimizes the predicted metric, bounded by a strict compute constraint.

### 2.3 Divergences from Standard MBO

While structurally isomorphic, we distinguish the neural scaling paradigm from standard MBO through three unique characteristics across its three stages:

1. **Low-Dimensional Spaces:** Standard MBO frequently navigates high-dimensional, unstructured, or highly rugged design spaces (e.g., continuous control parameters for robotics or combinatorial molecular structures). In contrast, the neural scaling paradigm operates within a relatively low-dimensional and macroscopic design space. The primary design variables—such as parameter count $N$, token count $D$, and data mixture $\boldsymbol{\omega}$—capture high-level structural properties. This compactness drastically reduces the search dimensionality and naturally constrains the complexity of the objective landscape.

2. **Strong Power-Law Priors:** Standard MBO typically employs "universal approximator" surrogates—such as Gaussian Processes (MacKay et al., 1998) or high-capacity MLPs—to model unknown black-box landscapes, relying heavily on first-order optimizers (e.g., Adam (Kingma & Ba, 2015)) to iteratively fit the surrogate over complex offline datasets. Conversely, neural scaling laws are grounded in strong power-law priors. Because the design space is low-dimensional and the surrogate acts as a rigid, transparent parametric function with minimal parameters, it bypasses the need for deep network fitting. This specific combination of a low-dimensional space and a strong functional prior allows researchers to leverage high-order optimization algorithms (such as L-BFGS (Liu & Nocedal, 1989)) to fit the surrogate, resulting in a highly robust, compute-efficient, and analytically tractable process.

3. **Strict Compute Constraints:** In standard MBO, the guided optimization stage typically seeks an unconstrained global or local optimum, bounded only by the validity of the design space. In contrast, guided optimization in the neural scaling paradigm is governed by strict compute budgets. The objective is to navigate the design space to identify the compute-optimal frontier—solving for the pre-training configuration $\boldsymbol{x}^*$ that optimizes the predicted validation metric subject to a hard compute constraint.

**Limitation** A standard MBO workflow often assumes a single unified surrogate that maps designs to objective values; in our setting, the natural analogue would be a unified scaling-law formula $\mathcal{L}(\boldsymbol{x}_m, \boldsymbol{x}_d, \boldsymbol{x}_h)$ that holds across all architectures and modalities. The principal limitation of the MBO framing is that we currently do not have a unified functional form: different architectural and modality choices induce fundamentally different surrogate forms — the Chinchilla form $\mathcal{L} = E + A/N^\alpha + B/D^\beta$ for dense transformers (Hoffmann et al., 2022), the bilinear MoE law of Clark et al. (2022) for routed models, and the multimodal laws of Shukor et al. (2025) all describe different parametric families that cannot be easily folded into a single closed expression (see Section 3 for the full survey).

## 3 The Design Space

We systematically categorize the design space $\mathcal{X}$ of neural scaling into three subspaces: **Model** ($\mathcal{X}_m$), **Data** ($\mathcal{X}_d$), and **Hyperparameters** ($\mathcal{X}_h$).

**Bi-level Optimization** Crucially, these subspaces do not occupy a parallel hierarchy. While a general MBO perspective considers all three subspaces as joint determinants of the objective, standard scaling laws treat the hyperparameter space $\mathcal{X}_h$ implicitly. They operate under the assumption that for any given model $\boldsymbol{x}_m$ and data $\boldsymbol{x}_d$, the hyperparameters have been tuned to their optimal configuration $\boldsymbol{x}_h^*(\boldsymbol{x}_m, \boldsymbol{x}_d)$.

We formalize this relationship as a bi-level optimization problem. The upper level optimizes model and data design variables subject to a strict compute budget, while the lower level resolves the optimal hyperparameters to ensure convergence for specific model and data configurations. Therefore, standard scaling laws explicitly model the validation metric $\mathcal{L}$ as a function of model and data only:

$$\begin{aligned}
\underset{\boldsymbol{x}_m \in \mathcal{X}_m, \, \boldsymbol{x}_d \in \mathcal{X}_d}{\text{minimize}} \quad & \mathcal{L}\big(\boldsymbol{x}_m, \boldsymbol{x}_d, \boldsymbol{x}_h^*(\boldsymbol{x}_m, \boldsymbol{x}_d)\big) \quad \text{subject to} \quad \text{Cost}(\boldsymbol{x}_m, \boldsymbol{x}_d) \leq C_{\text{budget}} \\
\text{where} \quad & \boldsymbol{x}_h^*(\boldsymbol{x}_m, \boldsymbol{x}_d) = \underset{\boldsymbol{x}_h \in \mathcal{X}_h}{\arg\min} \, \mathcal{L}(\boldsymbol{x}_m, \boldsymbol{x}_d, \boldsymbol{x}_h)
\end{aligned} \tag{2}$$

**Function Form** The most prominent formulation, the Chinchilla scaling law (Hoffmann et al., 2022), models $\mathcal{L}$ as a separable function of parameter count $N$ and training tokens $D$. It posits a parametric power-law of the form:

$$\mathcal{L}(N, D) = E + \frac{A}{N^\alpha} + \frac{B}{D^\beta} \tag{3}$$

where $E$ represents the irreducible loss (Bayes error), and $A, B, \alpha, \beta$ are fitted constants. This separable form assumes that the contributions of model scale and data scale are independent. In contrast, Kaplan et al. (2020) propose a coupled formulation:

$$\mathcal{L}(N, D) = \left[ \left( \frac{N_c}{N} \right)^{\frac{\alpha_N}{\alpha_D}} + \frac{D_c}{D} \right]^{\alpha_D} \tag{4}$$

While Eq. 4 is motivated by the intuition that $\mathcal{L}$ should follow individual power laws with respect to $N$ and $D$, it lacks an explicit irreducible loss term and couples the two variables, complicating the isolation of distinct scaling factors. Consequently, the separable form (Eq. 3) has become the standard for analyzing the comprehensive design space, which extends beyond $N$ and $D$ as detailed below.

For isolated analysis, the one-dimensional counterpart to Eq. 3 is the standard power law $\mathcal{L}(x) = E + \beta x^{-c}$, where $x$ represents data scale, model scale, or total compute. While widely adopted, this standard form fails to capture extreme boundary conditions. As $x \to \infty$, it cannot natively represent variance-limited behavior ($1/x$ scaling), prompting modified forms like $\mathcal{L}(x) = \beta(x^{-1} + \gamma)^{-c}$ (Bansal et al., 2022). Conversely, as $x \to 0$, it diverges to infinity rather than plateauing at random guessing performance $\mathcal{L}_0$. To correct this low-end saturation, researchers introduce bounded estimators, such as adding a translation parameter $d_{\text{sat}}$ (Zhai et al., 2022) or employing a sigmoidal transformation (Alabdulmohsin et al., 2022):

$$\frac{\mathcal{L}(x) - E}{(\mathcal{L}_0 - \mathcal{L}(x))^\lambda} = \beta x^{-c} \tag{5}$$

Despite these theoretical refinements, modern LLM pre-training typically operates either near the compute-optimal frontier or in over-trained regimes where both $N$ and $D$ are sufficiently large. In such high-compute environments, the standard power law remains practically sufficient.

## 3.1 Model Subspace ($\mathcal{X}_m$)

We examine the primary dimensions of the model subspace $\mathcal{X}_m$ and their influence on the validation metric. We partition this subspace into two orthogonal design vectors: *topology*, which dictates the structural graph and routing decisions, and *scale*, which governs the magnitude and precision.

### 3.1.1 Topology

**Dense Architecture** While we predominantly study the standard Transformer (Vaswani et al., 2017), the fundamental architectural design represents a pivotal discrete design variable within $\mathcal{X}_m$. Different architectures impose distinct inductive biases that directly modulate the scaling surrogate's parameters, specifically the exponent $\alpha$. Although $\alpha$ is theoretically constrained by the intrinsic dimension of the data manifold (Bahri et al., 2024), empirical evidence demonstrates that the realized scaling trajectory remains highly architecture-dependent (Tay et al., 2023). For instance, the Vanilla Transformer effectively maximizes spectral decay ($\alpha \approx 0.54$), whereas architectures with more restrictive inductive biases—such as convolutions (Wu et al., 2019) (LConv, $\alpha \approx 0.32$) or linear-attention mechanisms (Choromanski et al., 2021) (Performers, $\alpha \approx 0.25$)—introduce structural bottlenecks that significantly flatten the scaling curve. Furthermore, while scaling depth versus width yields a comparable validation metric for a fixed compute budget, downstream evaluations disproportionately favor deeper topologies.

**Mixture-of-Experts** Sparse Mixture-of-Experts (MoE) architectures extend structural topology by enabling conditional routing. Clark et al. (2022) propose a bilinear scaling law for validation loss that models the interaction between the base active parameter count $N$ and the effective expert count $\hat{n}_e$:

$$\ln \mathcal{L}(N, \hat{n}_e) \approx a_{\text{moe}} \ln N + b_{\text{moe}} \ln \hat{n}_e + c_{\text{moe}} \ln N \ln \hat{n}_e + d_{\text{moe}} \tag{6}$$

The interaction term ($c_{\text{moe}} > 0$) indicates that the marginal benefit of adding experts diminishes as the base model size $N$ increases. To unify the MBO design space across routed and dense models, this scaling law maps the MoE configuration to an *Effective Parameter Count* (EPC), $\bar{N}$, allowing direct performance comparisons. Further refining this topological design vector, Ludziejewski et al. (2024) identify *expert granularity* ($G_{\text{moe}}$) as a fundamental routing design variable. They demonstrate that optimizing $G_{\text{moe}}$ prevents the saturation of MoE benefits at larger scales, yielding a joint scaling law $\mathcal{L}(N, D, G_{\text{moe}})$ that strictly outperforms standard fixed-size expert configurations. Extending this, Zhao et al. (2025) formulate the ratio of shared to active experts ($S$) as another scale-independent design variable essential for stable specialized knowledge capture.

**Multimodal Fusion**  Integrating multiple modalities introduces a fundamental topological choice: early versus late fusion. Late fusion processes specialized data through dedicated encoders before backbone integration, whereas early fusion maps all modalities into a unified token stream. For compute-optimal scaling, Shukor et al. (2025) show that early-fusion architectures eliminate the parameter redundancy inherent in late-fusion vision encoders. Consequently, early-fusion operates efficiently at a lower optimal parameter-to-data ratio ($\frac{N}{D} \propto C^{0.053}$) than late-fusion ($\frac{N}{D} \propto C^{0.076}$). Furthermore, applying MoE to early-fusion models leverages *implicit modality specialization* via modality-agnostic routing.

Multimodal scaling introduces nonlinear interactions; as an early fusion framework, Aghajanyan et al. (2023) model this joint validation loss via constant synergy and scale-dependent functional competition:

$$\mathcal{L}(N, D_i, D_j) = \frac{\mathcal{L}(N, D_i) + \mathcal{L}(N, D_j)}{2} - \underbrace{\mathcal{C}_{i,j}}_{\text{Synergy}} + \underbrace{\left( \frac{A_{i,j}}{N^{\alpha_{i,j}}} + \frac{B_{i,j}}{(D_i + D_j)^{\beta_{i,j}}} \right)}_{\text{Competition}} \tag{7}$$

Here, $\mathcal{C}_{i,j}$ bounds the cross-modal information sharing, while the competition term quantifies degradation from modalities competing for finite parameter capacity $N$. This defines a structural phase transition: multimodal training incurs negative transfer at small scales, becoming beneficial only when $N$ crosses a threshold where synergy outweighs the competition penalty.

### 3.1.2 Scale

**Parameter Count**  The fundamental design variable dictating model scale is the parameter count $N$. However, defining this variable requires a nuanced distinction between *embedding* and *non-embedding* parameters. Kaplan et al. (2020) exclude embedding parameters to derive a compute-optimal configuration of $N_{\text{opt}} \propto C^{0.73}$. Conversely, Hoffmann et al. (2022) incorporate the total parameter count, resulting in a distinct relationship of $N_{\text{opt}} \propto C^{0.50}$:

$$N_{\text{opt}}(C) = G \left( \frac{C}{6} \right)^a, \quad \text{where} \quad G = \left( \frac{\alpha A}{\beta B} \right)^{\frac{1}{\alpha+\beta}}, \quad a = \frac{\beta}{\alpha + \beta}. \tag{8}$$

Recent analysis by Pearce & Song (2024) reconciles this discrepancy, demonstrating that the divergence is largely an artifact of counting methodologies; simulating the Chinchilla study using only non-embedding parameters at smaller scales reproduces coefficients closer to Kaplan's. To bypass these historical ambiguities, Bi et al. (2024) propose adopting *FLOPs per token* as a more rigorous design metric for scaling analysis.

Beyond the aggregate count, the parameter budget is distributed across specific structural configurations: vocabulary size ($V$), depth ($L$), and width ($d$). For a standard Transformer, the total count is approximated as $N \approx 12Ld^2 + Vd$, where $12Ld^2$ represents the $L$ layers (comprising $4d^2$ for attention and $8d^2$ for the feed-forward network) and $Vd$ accounts for the decoder layer. Width and depth typically scale synchronously ($d, L \propto N^{1/3}$) according to early paradigms (Kaplan et al., 2020). However, while these earlier studies often fixed $V$ at a constant size (e.g., 32K), Tao et al. (2024) demonstrate that maximizing information density requires scaling $V$ systematically with the model size to improve downstream performance. Under compute-optimal settings, they establish that optimal vocabulary parameters $N_v$ follow a power-law relationship with non-embedding parameters ($N_{nv}$), specifically $N_v \propto N_{nv}^{0.83}$. This leads to a refined, vocabulary-dependent surrogate formulation:

$$\mathcal{L}_u = -E + \frac{A_1}{N_{nv}^{\alpha_1}} + \frac{A_2}{N_v^{\alpha_2}} + \frac{B}{D^\beta} \tag{9}$$

where $\mathcal{L}_u$ is the unigram-normalized validation loss, ensuring equitable comparisons across varying tokenizers.

**Redundancy**   Standard power-law surrogates typically imply that the predicted validation loss decreases monotonically as the design variable $N$ increases. However, this functional form fails to capture capacity saturation in data-constrained regimes. Muennighoff et al. (2023) formalize this phenomenon through *model repetition*. When $N$ exceeds the optimal capacity $U_N$ warranted by a fixed dataset $D$, the excess parameters capture duplicative features rather than novel information. To integrate this constraint into the scaling surrogate, the raw parameter count $N$ is mapped to an *effective parameter count $N_{eff}$* via an exponential decay function governed by the repetition ratio $R_N$:

$$N_{eff} = U_N + U_N \cdot R_N^* \cdot \left( 1 - \exp \left( -\frac{R_N}{R_N^*} \right) \right) \tag{10}$$

Substituting $N_{eff}$ back into the standard power law accurately models the asymptotic validation plateaus characteristic of over-parameterized training. Consequently, an optimal design configuration must restrict this redundancy by maintaining a sufficiently high token-to-parameter ratio.

**Precision**   Standard surrogate modeling implicitly assumes high-precision arithmetic, treating precision as a fixed constant. Kumar et al. (2024) incorporate bit-width as an explicit capacity design variable, demonstrating that reducing precision mimics a reduction in parameter count. In this *precision-aware* scaling formulation, the effective parameter count $N_{\text{eff}}$ is defined as a multiplicative decay of the true parameters, governed by the bit-precision design $P$ of weights ($w$), activations ($a$), and KV cache ($kv$):

$$N_{\text{eff}} = N \cdot \prod_{x \in \{w, a, kv\}} \left( 1 - \exp \left( -\frac{P_x}{\gamma_x} \right) \right) \tag{11}$$

Furthermore, they model the degradation from post-training quantization (PTQ) as an additive penalty to the surrogate, $\delta_{\text{PTQ}}$, which paradoxically increases with the pre-training data volume $D$ (overtraining sensitivity). Combining these effects yields a unified scaling surrogate across both training and inference quantization regimes:

$$\mathcal{L}(N, D, P) = E + \frac{A}{(N_{\text{eff}})^\alpha} + \frac{B}{D^\beta} + \delta_{\text{PTQ}}(N_{\text{eff}}, D, P_{\text{train}}, P_{\text{post}}) \tag{12}$$

This formulation highlights a critical insight: while training in lower precision reduces $N_{\text{eff}}$, it acts as a form of "robustification," reducing the degradation $\delta_{\text{PTQ}}$ incurred when quantizing for inference.

### 3.2   Data Subspace ($\mathcal{X}_d$)

We examine the primary dimensions of the data subspace $\mathcal{X}_d$ and their influence on validation metrics. Paralleling the topological and scale vectors of the model subspace, we partition $\mathcal{X}_d$ into two orthogonal design vectors: *composition*, which defines the representational mapping and the domain distribution, and *scale*, which quantifies the absolute token volume and its effective scale.

#### 3.2.1   Composition

**Tokenizer**   The tokenizer functions as the compression interface between raw information (bytes) and the model's discrete input space (Delétang et al., 2024). Modern tokenizers are typically trained via algorithms such as Byte-Pair Encoding (BPE) (Sennrich et al., 2016) or Unigram language modeling (Kudo, 2018) to maximize the compression rate of the training corpus. This choice fundamentally redefines the data scale variable $D$: a more efficient tokenizer packs denser semantic content into fewer tokens, thereby increasing the effective context length. To maintain a consistent validation metric across different tokenizers, the standard cross-entropy loss $\mathcal{L}$ must be normalized by the compression ratio $\tau$ to obtain $\mathcal{L}_{\text{byte}} = \mathcal{L}/\tau$ (Hoffmann et al., 2022). While expanding the vocabulary size $V$ generally improves $\tau$ and reduces sequence length, it introduces a trade-off by increasing the parameter overhead of the embedding layer. The vocabulary size $V$ should not be treated as a static prior, but as a compute-dependent design variable co-optimized with model capacity ($N_v \propto N_{nv}^{0.83}$) to maximize information density per FLOP (Tao et al., 2024).

**Data Mixture** The data mixture represents a high-dimensional design variable within the data composition subspace. The training distribution is inherently heterogeneous, comprising diverse domains such as code, math, and wikipedia. Determining the optimal mixture vector $\boldsymbol{\omega}$ to balance these distributions is critical for downstream performance. Traditional methods like DoReMi (Xie et al., 2023) utilize a small proxy model to automatically determine optimal domain weights by solving a minimax objective to minimize the worst-case excess loss relative to a reference model:

$$\min_{\boldsymbol{\theta}} \max_{\boldsymbol{\omega} \in \Delta^k} \sum_{i=1}^{k} \boldsymbol{\omega}_i \left( \mathbb{E}_{x \in D_i}[\ell_{\boldsymbol{\theta}}(\boldsymbol{x}) - \ell_{\mathrm{ref}}(\boldsymbol{x})] \right) \tag{13}$$

However, a key limitation is that mixtures optimized at small proxy scales often fail to generalize to larger models. Consequently, recent methods have explicitly turned to model-based optimization, where a surrogate model is constructed to predict validation metrics based on the input mixture at scale.

Approaches like RegMix (Liu et al., 2024) circumvent the need for large-scale validation via the *rank invariance hypothesis*, fitting a regression model (e.g., LightGBM) on hundreds of ultra-small proxies to map mixture proportions $\boldsymbol{\omega}$ directly to the validation loss. Conversely, Ye et al. (2024) challenge this hypothesis, proposing *data mixing laws* that first extrapolate performance to larger scales, and subsequently fit an exponential functional form with respect to $\boldsymbol{\omega}$:

$$\mathcal{L}_i(\boldsymbol{\omega}) = c_{\mathrm{mix},i} + k_{\mathrm{mix},i} \exp \left( \sum_{j=1}^{K} t_{\mathrm{mix},ij} \boldsymbol{\omega}_j \right) \tag{14}$$

A similar surrogate strategy is explored by Dubey et al. (2024), employing a two-step fitting process: a power law mapping the compute budget to benchmark cross-entropy loss, followed by a sigmoid law mapping that loss to hard benchmark metrics. Most recently, Shukor et al. (2025) explore a unified surrogate function to predict validation loss by jointly modeling parameter count $N$, token count $D$, and the mixture design $\boldsymbol{\omega}$. They formalize an *additive law* (where mixture contribution is a separable term independent of scale), and a *joint law* (which captures the interaction between scale and data distribution):

$$\mathcal{L}(N, D, \boldsymbol{\omega}) = E + \frac{1}{\sum_{i=1}^{k} C_i \boldsymbol{\omega}_i^{\gamma_i}} + \frac{A(\boldsymbol{\omega})}{N^{\alpha}} + \frac{B(\boldsymbol{\omega})}{D^{\beta}} \tag{15}$$

where $A(\boldsymbol{\omega}) = \sum \boldsymbol{\omega}_i A_i$ and $B(\boldsymbol{\omega}) = \sum \boldsymbol{\omega}_i B_i$. While the aforementioned frameworks assume a static data mixture configuration, recent studies explore dynamic curriculum learning or mid-training phases (Olmo et al., 2025; Dubey et al., 2024). Nevertheless, due to the inherent complexity of large-scale pre-training, static mixtures remain the primary standard.

Multi-modality represents a highly complex extension of the data mixture. It extends beyond a single data type (e.g., text) to encompass distinct modalities such as images. This fundamentally differs from training on heterogeneous text sources (e.g., math and wikipedia), which, despite their domain differences, share underlying linguistic patterns and a common modality space. In the multimodal setting, the token count $D$ assumes a dual role: it provides information gain while simultaneously introducing *optimization competition* between modalities $D_i$ and $D_j$. Aghajanyan et al. (2023) model this phenomenon via the data-dependent interaction term $\frac{B_{i,j}}{(D_i + D_j)^{\beta_{i,j}}}$. In contrast, Shukor et al. (2025) do not explicitly separate architecture or data terms by modality. Instead, they treat all text and image inputs as a unified stream of tokens governed by specific mixture. Consequently, the final token composition is tied to this mixture design. For instance, in their default setup (45% image-caption, 45% interleaved, and 10% text), vision tokens ($D_v$) account for 54.4% of the total training tokens, driven by the high visual token density inherent in captioning datasets.

### 3.2.2 Scale

**Token Count** The design variable $D$ represents the absolute token count processed during optimization. The theoretical understanding of $D$'s role within the data subspace has shifted significantly. Early work by Kaplan et al. (2020) posited that performance is weakly dependent on $D$, suggesting that the compute

budget should be allocated disproportionately toward the model size $N$ (scaling $N \propto C^{0.73}$). However, the seminal *Chinchilla* analysis (Hoffmann et al., 2022) corrected this misconception, demonstrating that $N$ and $D$ contribute symmetrically to the reduction of validation loss.

$$D_{opt}(C) = G^{-1}\left(\frac{C}{6}\right)^b, \quad \text{where} \quad G = \left(\frac{\alpha A}{\beta B}\right)^{\frac{1}{\alpha+\beta}}, \quad b = \frac{\alpha}{\alpha+\beta}. \tag{16}$$

Under the compute-optimal allocation, the model parameter count and training token count should scale linearly with each other ($N \propto D$). This implies that data is not merely a consumable resource, but a fundamental design dimension as important as the model scale.

**Repetition** Managing data repetition involves a delicate tension between mitigating harmful redundancy and addressing data scarcity. To ensure the training set $\mathcal{D}$ contains high-signal information, modern preparation pipelines (Olmo et al., 2025) treat redundancy as noise and typically employ a cascade of deduplication strategies: *exact deduplication* to remove identical documents, *fuzzy deduplication* (e.g., MinHash LSH) to identify near-duplicates, and *substring deduplication* to excise repeated boilerplate text.

However, in data-constrained regimes where unique high-quality tokens are exhausted, treating repetition as an explicit design variable becomes necessary. Muennighoff et al. (2023) observe that when a fixed dataset of $U_D$ unique tokens is repeated over multiple epochs, the marginal contribution of each additional token decays. To formalize this, they define the repetition factor $R_D$ as:

$$R_D = \max\left(0, \frac{D}{U_D} - 1\right) \tag{17}$$

The authors propose that repeated tokens contribute to an *effective data scale $D_{eff}$* rather than the raw token count $D$. This is modeled using an exponential decay function:

$$D_{eff} = U_D + U_D \cdot R_D^* \cdot \left(1 - \exp\left(-\frac{R_D}{R_D^*}\right)\right) \tag{18}$$

where $R_D^*$ is a learned constant representing the *half-life* of the value of repeated data.

Recent work expands repetition to encompass *semantic redundancy*. Chen et al. (2025) demonstrate that semantically similar, yet token-unique, documents effectively act as repetition, reducing information gain. This redundancy is fundamentally scale-dependent. Sorscher et al. (2022) identify a phase transition in optimization: while data-scarce regimes require "easy" prototypical data (near cluster centroids), these samples become redundant as the data scale increases. Thus, data-abundant configurations must shift sampling toward "hard" outliers to maximize information gain. Formalizing this dynamic, Covert et al. (2024) model the marginal value $\psi_k(z)$ of adding a data point $z$ to a set of size $k$ via a power-law decay:

$$\psi_k(z) \approx c(z)k^{-\alpha(z)} \tag{19}$$

where $c(z)$ is the initial information magnitude and $\alpha(z)$ represents the redundancy rate. This unifies empirical observations: "easy" samples exhibit high $c(z)$ but rapid decay (large $\alpha(z)$), whereas "hard" samples decay slower (small $\alpha(z)$). Consequently, optimal data selection necessitates accounting for *ranking reversals*, as the most valuable samples at small scales differ fundamentally from those required for large-scale configurations.

Finally, Sardana et al. (2024) identify a distinct form of *implicit repetition* arising from model capacity constraints rather than data distribution. When the parameter count $N$ is fixed, the marginal benefit of adding entirely novel data diminishes rapidly, causing $D_{eff}$ to plateau. Crucially, this mechanism is symmetric to the model repetition formalized in Eq. 10: just as excess parameters redundantly encode a fully resolved data manifold, excess unique data becomes functionally redundant when the model's representational capacity saturates. These phenomena collectively reveal that the standard separable Chinchilla scaling surrogate is merely a macroscopic approximation. By treating $N$ and $D$ as independent terms, it fundamentally fails to account for the mutually restrictive capacity bottlenecks between the model and data scale designs.

**Quality**   Beyond repetition, *data quality* serves as a critical design variable governing the effective data scale ($D_{eff}$). Tokens are not fungible with respect to optimizing validation metrics; for instance, *textbook-quality* data (Gunasekar et al., 2023) yields a significantly steeper descent in the loss landscape compared to noisy web text. This phenomenon was empirically demonstrated by the Phi-1 model, which, by pre-training on merely 7B tokens of curated textbook-quality data (8 epochs) and fine-tuning on 200M tokens of exercises, outperformed substantially larger competitors on HumanEval and MBPP benchmarks.

To operationalize this, modern pipelines employ learned classifiers to filter training distributions. A standard approach involves training a fastText-based classifier to distinguish high-quality text from low-quality noise (Li et al., 2024). By assigning a quality score $q_i$ to each document and filtering via thresholding or importance sampling, one effectively optimizes the data scaling coefficient within the surrogate model, allowing models to achieve target validation metrics with a significantly reduced data budget.

Formalizing this intuition, Subramanyam et al. (2025) introduce a *quality-aware scaling law*, modeling the effective data scale as $D_{eff} = D \cdot g(Q)$. Here, $Q \in (0, 1]$ is a dimensionless parameter quantifying data corruption, and the scaling function is modeled as a power law $g(Q) = Q^\gamma$, where $\gamma$ is a learned parameter. Similarly, in the context of neural machine translation, Bansal et al. (2022) investigate the scaling properties of dependent noise by generating back-translated data via reverse models. They report that such synthetic data exhibits a significantly lower scaling exponent ($\beta \approx 0.19$) compared to organic parallel data ($\beta \approx 0.28$), indicating that the marginal quality of synthetic data diminishes much faster than that of real data in large-scale regimes.

### 3.3   Hyperparameter Subspace ($\mathcal{X}_h$)

This subspace encompasses the training hyperparameters required to ensure the model converges to its minimal potential loss $\mathcal{L}^*$, conditioned on specific model and data configurations:

$$\boldsymbol{x}_h^*(\boldsymbol{x}_m, \boldsymbol{x}_d) = \underset{\boldsymbol{x}_h \in \mathcal{X}_h}{\arg\min} \, \mathcal{L}(\boldsymbol{x}_m, \boldsymbol{x}_d, \boldsymbol{x}_h) \tag{20}$$

**Joint Optimization**   Assuming the model architecture family and training data distribution remain constant, the relevant components of $\boldsymbol{x}_m$ reduce to the parameter count ($N$), and $\boldsymbol{x}_d$ reduces to the token count ($D$). Consequently, the core hyperparameter design task simplifies to determining the optimal batch size and learning rate pair ($B_s, \eta$) given $N$ and $D$. Because brute-force grid searches scale intractably with $\mathcal{O}(|\mathcal{B}| \cdot |\mathcal{H}| \cdot |\mathcal{N}| \cdot |\mathcal{D}|)$, researchers must construct predictive surrogate models to map scale to optimal hyperparameter configurations. Crucially, this differs from the upper-level MBO surrogate: rather than predicting validation metrics to optimize model and data configurations, it directly outputs the optimal hyperparameters as a function of scale.

#### 3.3.1   Batch Size

We prioritize batch size ($B_s$) as a primary design variable because it serves as the critical bridge between convergence dynamics and hardware system throughput. Prior studies might fix $B_s$ to be sufficiently large to isolate the learning rate (Zhou et al., 2026), but optimizing it directly is essential for compute efficiency.

**Critical Batch Size**   The optimal configuration of $B_s$ is fundamentally governed by the *Critical Batch Size* ($B_{s,\mathrm{crit}}$). Kaplan et al. (2020) formalized this via the *Gradient Noise Scale* (GNS), defined as the ratio of single-sample gradient variance to the squared magnitude of the true global gradient. For any batch size, the Signal-to-Noise Ratio (SNR) of the update is $B_s/\mathrm{GNS}$.

The critical batch size is the equilibrium point ($B_s \approx \mathrm{GNS}$) where $\mathrm{SNR} \approx 1$. If $B_s \ll B_{s,\mathrm{crit}}$, the gradients become highly inaccurate (low SNR), forcing the optimizer to take an excessive number of sequential steps to converge, which shatters hardware utilization. Conversely, in the signal-dominated regime ($B_s \gg B_{s,\mathrm{crit}}$), the model suffers from "step starvation"; massive batches yield severely diminishing returns per update, wasting raw compute. Crucially, $B_{s,\mathrm{crit}}$ is dynamic, exhibiting a power-law growth as loss decreases ($B_{s,\mathrm{crit}} \propto \mathcal{L}^{-1/\alpha_B}$). However, because dynamically resizing $B_s$ during training necessitates costly reconfigurations of 3D parallelism strategies, standard practice typically seeks an optimal static approximation for the entire run.

**Compute-Optimal Fit**  Modeling $B_s$ across the fully decoupled design space of $(N, D)$ is highly complex; therefore, practical design strategies typically focus on the compute-optimal frontier. Bi et al. (2024) operationalize this by evaluating various learning rate and batch size combinations to model how optimal batch size scales with the total compute budget $C$. Through empirical grid searches spanning $10^{17}$ to $2 \times 10^{19}$ FLOPs, they derive a robust power law:

$$B_{s,\text{opt}}(C) \approx 0.2920 \cdot C^{0.3271} \tag{21}$$

This empirical fit ($B_{s,\text{opt}} \propto C^{0.33}$) acts as a practical pre-determination of the critical batch size for large-scale runs. Theoretically, it perfectly aligns with Chinchilla scaling limits (Hoffmann et al., 2022): substituting the macroscopic loss trajectory $\mathcal{L}(C) \approx a_{\text{bs}} C^{-b_{\text{bs}}}$ (assuming irreducible loss $E \approx 0$) into the critical batch size relation $B_s \propto \mathcal{L}^{-1/\alpha_B}$ analytically yields $B_s \propto C^{b_{\text{bs}}/\alpha_B}$. Similarly, Porian et al. (2024) model the optimal batch size strictly as a function of parameter count $N$ and they employ a rigorous two-stage methodology: first, for a fixed $N$ and $B_s$, they apply Akima interpolation across the learning rate domain to identify the minimal achievable loss; second, holding $N$ constant, they interpolate across these minimized loss values to pinpoint the global optimal batch size ($B_s^*$). This yields the scaling law:

$$B_{s,\text{opt}}(N) = 0.7576 N^{0.703} \tag{22}$$

Notably, these distinct surrogate formulations are mathematically consistent under compute-optimal assumptions. Given the standard token allocation $D \approx 20N$, total compute scales as $C \approx 6ND \approx 120N^2$, implying $N \propto C^{1/2}$. Substituting this into Porian's surrogate yields $B_{s,\text{opt}} \propto (C^{1/2})^{0.703} \approx C^{0.3515}$, which tightly mirrors the empirical $C^{0.3271}$ exponent derived by Bi et al. (2024).

### 3.3.2 Learning Rate

**Schedule**  To model the learning rate $\eta$, researchers must first define the optimization schedule. The warmup-stable-decay (WSD) schedule has recently superseded traditional Cosine decay as the industry standard (Hu et al., 2024; Hägele et al., 2024). Cosine decay inherently couples the learning rate trajectory to the token count ($D$), meaning a run is only optimal at its exact final step. By contrast, the extended constant phase of WSD decouples the optimization path from the total compute budget. Applying stochastic weight averaging (SWA) (Izmailov et al., 2018) to merge checkpoints around a target evaluation step allows this single WSD run to recover performance comparable to a fully optimized, budget-specific cosine decay. In an MBO context, this decoupling is invaluable: it effectively eliminates the need for computationally expensive, independent training runs by allowing a single WSD trajectory to evaluate multiple token budgets, thereby providing multiple design-label pairs efficiently.

**Parameterization**  The optimal learning rate is contingent not only on parameter count ($N$) but also on the training token count ($D$). While approaches like maximal update parametrization ($\mu$P) (Yang et al., 2022) theoretically enable the width-invariant transfer of optimal learning rates from small to large models, their practical integration can be complicated by this dependency on $D$ alongside the nuances of real-world training dynamics. Consequently, standard parameterization (SP) frequently remains the default choice in large-scale practice. The continued viability of SP at scale is largely attributed to the natural evolution of modern architectures. For instance, many of the early stability challenges associated with scaling—such as controlling activation scales—are now routinely mitigated through standard improvements like QK-Norm (Henry et al., 2020). Furthermore, as demonstrated by Everett et al. (2024), optimal learning rate transfer under SP can also be achieved through careful scale-wise and layer-wise multipliers.

**Compute-Optimal Fit**  Under standard compute-optimal settings (where $D \approx 20N$), researchers typically model the learning rate primarily as a function of either model scale or total compute, effectively collapsing the $(N, D)$ design space into a single dimension. Porian et al. (2024) derive fitting data points through a robust three-stage interpolation process over $N$, $B_s$, and $\eta$. First, for each fixed pair of $(N, B_s)$, Akima interpolation is applied to the loss landscape to identify the minimal loss. Second, holding $N$ constant, they interpolate across the resulting optimal loss-batch size pairs to determine the global optimal batch size $B_s^*$. Finally, utilizing this relationship, they interpolate to find the specific learning rate $\eta^*$ corresponding to $B_s^*$.

By performing linear regression on these identified optimal points in log-log space, they fit the final power-law:

$$\eta_{opt}(N) = 3.7N^{-0.36} \tag{23}$$

This inverse relationship dictates that larger models necessitate reduced learning rates to maintain numerical stability. Similarly, Bi et al. (2024) formulate the optimal learning rate as a function of the total compute budget $C$. By conducting extensive grid searches on compute budgets ranging from $10^{17}$ to $10^{19}$ FLOPs near the compute-optimal frontier, they identify:

$$\eta_{opt}(C) \propto C^{-0.125} \tag{24}$$

Notably, this result exhibits mathematical consistency with Eq. (23). Approximating the compute budget as $C \approx 6ND \approx 120N^2$ (implying $C \propto N^2$), DeepSeek's relationship transforms to $\eta_{opt} \propto (N^2)^{-0.125} = N^{-0.25}$. Both laws confirm that $\eta_{opt}$ follows a negative power law with respect to scale.

**Non-Compute-Optimal Fit**   In real-world scenarios, organizations frequently over-train smaller models beyond the Chinchilla optimal point to minimize inference costs. This necessitates a decoupled analysis of $N$ and $D$, which significantly expands the design space and makes full grid searches computationally prohibitive. To mitigate this $\mathcal{O}(|\mathcal{H}| \cdot |\mathcal{N}| \cdot |\mathcal{D}|)$ search complexity, Zhou et al. (2026) exploit geometric priors via a three-step process: (1) For a fixed $N$ and $D$, the loss $\mathcal{L}$ is modeled as a parabola in log-LR space ($\mathcal{L}(\eta) \approx \mathcal{L}_{\min} + C_{\mathrm{lr}}(\log \eta - \log \eta^*)^2$), minimizing the discrete evaluation points required. (2) *Trajectory Extrapolation* is employed, projecting short-run loss curves to the target data scale using a power law $\mathcal{L}(D) \approx \mathcal{L}_0 + A_{\mathrm{lr}}D^{-\gamma_{\mathrm{lr}}}$. (3) The derived optimal points are aggregated to formulate the final scaling law:

$$\eta^*(N, D) = 38.4588 \cdot N^{-0.2219} \cdot D^{-0.3509} \tag{25}$$

Alternatively, this relationship can be expressed as $\eta^*(N, D) = 13.4420 \cdot N^{-0.5728} \cdot (D/20N)^{-0.3509}$, in which the $(D/20N)^{-0.3509}$ component functions as a correction term for deviations from the compute-optimal fit.

## 4   An Illustrative Proof-of-Concept: Autofocus for Scaling Law Extrapolation

In this section, we focus explicitly on the *surrogate modeling* stage of the MBO framework. We empirically demonstrate how an established MBO technique—*autofocus* (Fannjiang & Listgarten, 2020; Cleveland, 1979)—can enhance the extrapolation accuracy of scaling laws.

**Disclaimer:** The objective of this section is not to establish a new state-of-the-art scaling law functional form or to present a comprehensive empirical benchmark. Rather, it serves as an illustrative proof-of-concept to validate the conceptual mapping established in Section 2. We utilize a constrained dataset to demonstrate that adopting an MBO-native technique (autofocus) inherently resolves specific extrapolation pathologies found in standard surrogate fitting.

### 4.1   Motivation and the MBO Perspective

Within the neural scaling paradigm, the surrogate modeling stage typically relies on robust regression objectives, such as the Huber loss (Huber, 1992), to mitigate the influence of outliers. However, the standard practice of treating all data points as equally informative implicitly assumes that a parametric surrogate model—such as the Chinchilla functional form, $\mathcal{L}(N, D)$—holds perfectly across the entire design space.

In reality, these power laws serve only as global approximations. Local scaling behaviors often vary depending on the specific regime; for instance, the loss dynamics of overtrained models differ fundamentally from those of undertrained models. Viewed through the MBO lens, this discrepancy necessitates *autofocus*. Rather than fitting a single global surrogate, we should autofocus the surrogate to better capture the relevant scaling dynamics by assigning higher weights to points that share similar characteristics with the target configuration.

### 4.2   Proposed: Autofocused Scaling Laws

We propose *autofocused scaling laws*, a framework that dynamically adapts the parametric surrogate model by autofocusing it around the target configuration.

**Autofocus**  Autofocus models the distribution around the target configuration and proposes importance sampling to reweight the training loss. In standard practice, the offline dataset of training configurations is implicitly assumed to be drawn from a uniform base distribution, $p_0(\boldsymbol{x}) \propto 1$, meaning all configurations contribute equally to the empirical risk. However, predicting scaling behavior at an extrapolation configuration $(N_*, D_*)$ vastly different from the training data induces a severe covariate shift. To accurately predict this configuration, we want our surrogate to reflect a localized target generative distribution, $p_*(\boldsymbol{x})$, centered around this region of interest. The importance weight for each training configuration $\boldsymbol{x}_i$ is therefore governed by the density ratio $w_i \propto p_*(\boldsymbol{x}_i)/p_0(\boldsymbol{x}_i)$. By locking the surrogate's optimization to these importance weights, autofocus dynamically corrects for the extrapolation-induced covariate shift. Formally, the parameters of the surrogate model, $\boldsymbol{\theta}$, are estimated by minimizing the importance-weighted empirical risk:

$$\hat{\boldsymbol{\theta}} = \arg\min_{\boldsymbol{\theta}} \frac{1}{M} \sum_{i=1}^{M} \frac{p_*(\boldsymbol{x}_i)}{p_0(\boldsymbol{x}_i)} \cdot \ell(\boldsymbol{x}_i, \boldsymbol{\theta}) \tag{26}$$

**Generative Modeling**  When constructing the generative model for the target configuration, a critical design choice is the parameterization of the feature space. We advocate for a decoupled coordinate system: the log model scale, $\log N$, and the log data-to-parameter ratio, $\log r = \log(D/N)$. To ensure the distance metric remains consistent across any extrapolation setting, we apply min-max normalization, mapping all configurations into the interval $[0, 1]$ to yield $\tilde{N}$ and $\tilde{r}$. Assuming the base distribution $p_0(\boldsymbol{x})$ is uniform, the density ratio becomes directly proportional to the target density $p_*(\boldsymbol{x})$. We model $p_*(\boldsymbol{x})$ as an unnormalized Gaussian density centered at the normalized target configuration $(\tilde{N}_*, \tilde{r}_*)$, yielding the sample weight:

$$w_i' = \exp\left(-\frac{(\tilde{N}_i - \tilde{N}_*)^2}{2\sigma_N^2} - \frac{(\tilde{r}_i - \tilde{r}_*)^2}{2\sigma_r^2}\right) \tag{27}$$

In an ablation study, we test an alternative feature space using raw scale variables, $(\log N, \log D)$, and find it yields inferior performance, as detailed in Sec. 4.4.

**Final Objective**  To ensure gradient magnitudes remain consistent with the uniform baseline, we normalize the weights to have a mean of exactly 1 (i.e., $w_i = w_i'/(\frac{1}{M}\sum_{j=1}^{M} w_j')$). These normalized weights are then incorporated into the standard Huber loss objective, computed on the log-loss residuals:

$$\mathcal{J}(\boldsymbol{\theta}) = \sum_{i=1}^{M} w_i \cdot \text{Huber}_\delta\big(\log \hat{\mathcal{L}}_i(\boldsymbol{\theta}) - \log \mathcal{L}_i\big) \tag{28}$$

where we perform a hyperparameter search over $\delta \in \{10^{-1}, 10^{-2}, 10^{-3}\}$ and employ fixed default bandwidths of $\sigma_N = \sigma_r = 1.0$. We empirically evaluated alternative bandwidth hyperparameters, but found they yielded identical or marginally degraded performance, justifying our default settings. See Appendix A for implementation details and Appendix B for additional sensitivity analyses of Autofocus.

## 4.3  Experiment Design

To validate the extrapolation effectiveness of our proposed autofocused scaling laws, we conduct a series of empirical evaluations in this subsection. We leverage the public dataset provided by Porian et al. (2024), extracting 200 $(N, D, \mathcal{L})$ triplets across 16 distinct model scales. We define $N$ as the effective parameter count, derived directly from precise FLOPs-per-token to account for sequence-length-dependent operations. Our target evaluation set comprises the two largest model scales ($N = 7.43 \times 10^8$ and $1.08 \times 10^9$), yielding a total of 9 test configurations.

To assess performance under varying degrees of extrapolation difficulty, we construct four progressively challenging data availability regimes where the training configurations are restricted by an upper bound threshold on the parameter count. **Setting 1** includes all available configurations except for the target evaluation set, where the maximum training parameter count is $0.39\times$ the target parameter count. **Setting 2** restricts the training configurations to models with less than half the target parameter count ($N \leq 3.72 \times 10^8$).

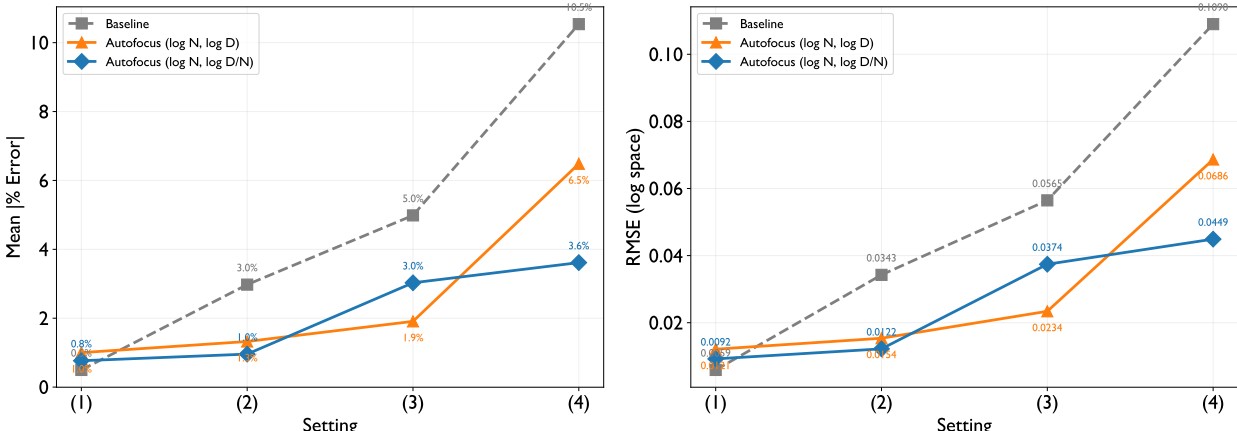

Figure 2: Mean absolute percentage error and RMSE (log space) across settings (1)-(4).

**Setting 3** restricts the configurations to one-quarter of the target parameter count ($N \leq 1.86 \times 10^8$). Finally, **Setting 4** restricts the configurations to one-eighth of the target parameter count ($N \leq 9.29 \times 10^7$).

To address the highly non-convex nature of fitting the 5-parameter parametric surrogate model, we perform the optimization using 15 distinct random initializations. The final predicted loss for any target configuration is then computed as the average of the log-loss predictions across these 15 fits.

### 4.4 Experimental Results and Analysis

Based on our empirical evaluation across these four settings (summarized in Figure 2 and Figure 3), we highlight three observations:

**(1) Performance in Data-Rich Regimes is Comparable:** In Setting (1), where the offline training configurations contain models very close to the target configurations, the baseline (0.50% error) and autofocus (0.77% error) perform similarly. In such interpolation-like regimes, a global uniform fit is generally sufficient, and local reweighting yields marginal differences.

**(2) Autofocus Excels in Data-Constrained Extrapolation:** As training configurations become severely limited, autofocus provides clear and substantial gains. In Setting (4), the uniform baseline error degrades sharply to 10.54%. In contrast, autofocus successfully localizes the fit to the boundary of the relevant offline configurations, reducing the error to 3.61%—a nearly 2.9× reduction.

**(3) Feature Space Ablation Highlights Complementary Strengths:** The $(\log N, \log D/N)$ space consistently wins in Settings 1, 2, and 4, confirming that the data-to-parameter ratio ($D/N$) is a fundamental proxy for scaling behavior. Regardless of the chosen feature space, autofocusing strictly outperforms the uniform baseline in all true extrapolation scenarios (Settings 2 through 4).

### 4.5 Discussion: Validating the MBO Paradigm

The empirical success of autofocusing directly substantiates the core thesis of this paper: fitting scaling laws is fundamentally a *surrogate modeling* problem within the broader MBO framework. When we rely solely on uniform fitting, we implicitly trust that our parametric surrogate model perfectly represents the true loss landscape across all scales. The rapid degradation of the baseline in Setting (4) demonstrates the danger of this assumption.

By formally adopting the MBO perspective, we recognize the scaling law not as an absolute ground truth, but as a localized surrogate model. Applying autofocus—a technique native to the MBO literature—allows us to dynamically correct for the distributional shift between the small-scale training configurations and the large-scale target configurations. While the 2.9× reduction in extrapolation error is empirically encouraging,

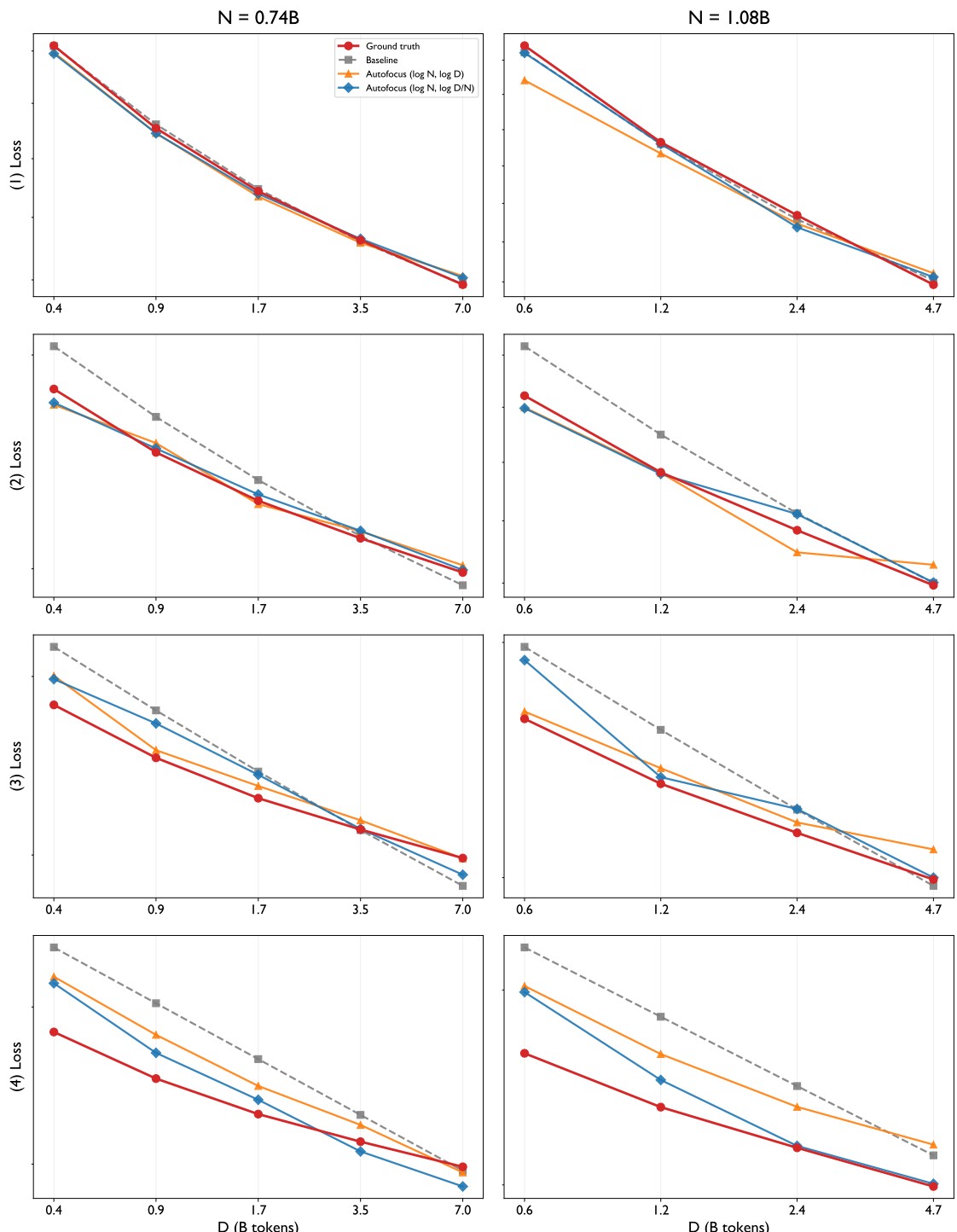

Figure 3: Surrogate model test predictions ($\mathcal{L}$ vs. $D$) for the 0.74B and 1.08B models across all settings.

**the primary takeaway is the value of this perspective shift**. This perspective shift becomes operative precisely because of neural scaling's strong functional prior. Standard MBO surrogates are universal approximators (e.g., MLPs); locally up-weighting points around a target with such a model simply specializes it to the local region and loses extrapolation. By contrast, the rigid 5-parameter power-law surrogate (Chinchilla form, Eq. 3) constrains the fit to a global functional form, so autofocus's local re-weighting only

re-positions the few free parameters to better capture the local regime — exactly the property that makes the MBO toolkit applicable here. The accuracy gains serve merely as a symptom of a deeper structural alignment: treating scaling laws as surrogate models successfully unlocks decades of established MBO methodologies. This confirms that our framework is not merely a theoretical relabeling, but a highly practical and necessary paradigm shift for navigating the strict compute constraints of foundation model pre-training.

**Limitations and Expected Behavior at Scale.** Scaling laws are approximations of the true loss landscape; autofocus's contribution is to upweight training points near the prediction target when fitting the surrogate. This principle does not depend on absolute scale, so we expect the same qualitative behavior at (i) larger model sizes and (ii) larger token budgets. Empirically validating this carry-over at frontier scales ($\geq$ 10B parameters, trillion-token budgets) would require pre-training compute we do not have access to, and no public dataset of comparable diversity exists at those scales. We therefore deliberately limit the empirical commitment of this position paper to a verifiable open dataset (Porian et al., 2024) and flag frontier-scale validation as the natural empirical follow-up.

## 5 Conclusion

### 5.1 Summary of Contributions

In summary, our primary contribution is formally bridging the gap between empirical neural scaling practices and MBO by mapping the scaling paradigm to the three stages of MBO: *design space*, *surrogate modeling*, and *guided optimization*. Within this framework, we systematically partition the scaling design space into *model*, *data*, and *hyperparameter* subspaces. Crucially, we unify these dimensions by formalizing their relationship as a bi-level optimization problem, wherein hyperparameters are optimized at the lower level to ensure convergence for specific model and data configurations. Finally, we demonstrate the practical utility of this conceptual mapping by applying *autofocus* to the surrogate modeling stage, providing an illustrative proof-of-concept that explicitly mitigates extrapolation-induced covariate shifts and minimizes prediction errors in data-constrained regimes.

### 5.2 Roadmap for Future Research

**Uncertainty Quantification** Standard parametric surrogate models yield deterministic point estimates $y = \hat{f}(\boldsymbol{x})$, ignoring the variance and confidence intervals that are crucial for high-stakes compute allocation. In industrial practice, justifying the expenditure of tens of millions of dollars requires more than a single extrapolated surrogate prediction; it demands rigorous risk assessment. To make the design process robust, we must quantify epistemic uncertainty. While bootstrapping provides a basic approximation, a more principled MBO approach is *Bayesian Regression*. By modeling the scaling coefficients (e.g., $\alpha, \beta$) not as fixed scalars but as random variables with specific priors, we can derive a full posterior distribution given the observed offline small-scale configurations. This enables risk-aware optimization—allowing decision-makers to maximize the lower confidence bound (LCB) of predicted performance to guarantee baseline capabilities before committing massive compute budgets. While recent work by Lee et al. (2025) explores predicting distributions using a transformer model, their approach is limited to single-variable settings; a parametric Bayesian approach naturally scales to the multi-variable $\mathcal{L}(N, D)$ setting required for realistic foundation model scaling. Crucially, quantifying this uncertainty unlocks active exploration. Prior approaches rely on static grid searches or heuristic selections of parameter count $N$ and token count $D$, often wasting compute on redundant experiments. By treating scaling exploration as a sequential MBO problem, researchers can leverage an acquisition function to dynamically determine the exact design variables $\boldsymbol{x}$ that maximize expected information gain. This transforms scaling analysis from a static surrogate modeling exercise into an active learning process that optimally directs limited R&D compute budgets.

**Multi-Objective Optimization** Recent scaling research has predominantly focused on optimizing a single, scalar validation metric. However, foundation models must increasingly satisfy a vector of competing objectives, $\boldsymbol{y} = [y_{\text{general}}, y_{\text{code}}, y_{\text{reasoning}}, y_{\text{multi-lingual}}]$, particularly during the critical phase of pre-training data mixture optimization. Standard practice typically reduces these diverse objectives to a weighted sum

via linear scalarization. This approach is fundamentally limited because the Pareto frontier of diverse model capabilities is often non-convex; a simple weighted average cannot recover optimal solutions located within the concave regions of the trade-off surface. To address this, we advocate for the adoption of formal multi-objective optimization techniques, such as Tchebycheff scalarization (Steuer & Choo, 1983), natively utilized in MBO frameworks. By applying these methods to data mixture optimization, researchers can explicitly navigate the complex trade-offs between pre-training domains. This rigorous approach ensures that prioritizing one capability (e.g., code) does not catastrophically degrade performance in another (e.g., multi-lingual) under a strict compute budget.

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

# A    Implementation Details

**Log-space parameterisation.**    We fit $\log \hat{\mathcal{L}} = \text{logsumexp}(\log E, \ \log A - \alpha \log N, \ \log B - \beta \log D)$ rather than $\hat{\mathcal{L}}$ directly, with optimisation variables $(\log E, \log A, \alpha, \log B, \beta)$.

**Optimiser.**    L-BFGS-B with `ftol=1e-15`, `gtol=1e-12`, `maxiter=5000`, over the bounds $\log E \in [\log 0.01, \log 10]$, $\log A, \log B \in [\log 0.01, \log 10^6]$, $\alpha, \beta \in [0.01, 2]$.

**15-restart ensemble and reproducibility.**    The 15 random restarts already specified in Section 4.3 draw from $E \sim \mathcal{U}(0.3, 3)$, $A, B \sim \mathcal{U}(1, 100)$, $\alpha, \beta \sim \mathcal{U}(0.1, 1)$ via a fixed `np.random.RandomState(42)`; all 15 converged solutions are retained and their log-predictions are averaged for reporting. To verify stability we re-ran the entire 15-restart ensemble pipeline under 5 independent master seeds; the mean absolute percentage error fluctuates by less than 0.5 pp across runs in every setting, so the reported numbers are reproducible up to negligible noise.

# B    Sensitivity Analyses for Autofocus

**Normalization**    We compare global min-max normalization, z-score normalization (zero mean, unit std), and no normalization (raw log-features), keeping the Gaussian kernel with $\sigma = (1, 1)$ and $\delta = 10^{-3}$.

Table 1: Normalization ablation.

| Setting | Baseline | min-max (paper) | z-score | none |
|---|---|---|---|---|
| (1) | 0.50% | **0.77%** | 1.33% | 0.88% |
| (2) | 2.98% | **0.96%** | 1.59% | 1.60% |
| (3) | 4.98% | 3.03% | 2.56% | **2.14%** |
| (4) | 10.54% | **3.61%** | 7.11% | 6.25% |

As shown in Table 1, min-max wins in three of four settings and is by far the best in the hardest setting (4), where the z-score and no-normalization variants degrade sharply (7.11%/6.25%). Min-max bounds both features into $[0, 1]$ so that a single default $\sigma$ has consistent meaning across settings; z-score's std-based scale shifts as the training pool shrinks, and the raw-log variant gives the two features very different ranges.

**Kernel Form**    We compare the Gaussian kernel (paper's choice) against the Epanechnikov kernel $K(u) = \max\left(\frac{3}{4}(1 - u^2), 0\right)$. Same $\sigma = 1$ in normalized $[0, 1]$ feature space.

Table 2: Kernel form ablation.

| Setting | Baseline | Gaussian (paper) | Epanechnikov |
|---|---|---|---|
| (1) | 0.50% | **0.77%** | 1.01% |
| (2) | 2.98% | **0.96%** | 1.06% |
| (3) | 4.98% | 3.03% | **2.68%** |
| (4) | 10.54% | **3.61%** | 4.34% |

As shown in Table 2, both local-kernel variants beat the uniform baseline in the data-constrained settings (2)–(4). Gaussian wins in (1), (2), and (4); Epanechnikov wins in (3). The choice of kernel matters less than the *fact* of localization.

**Bandwidth**    We sweep $\sigma_N, \sigma_{D/N} \in \{0.5, 1.0, 2.0\}$ keeping all other choices at the paper's defaults (min-max normalization, Gaussian kernel, $\delta = 10^{-3}$). Paper's default $\sigma = (1, 1)$ is bolded.

Table 3: Bandwidth ablation.

| **Setting (1)**, baseline 0.50% | | | | **Setting (2)**, baseline 2.98% | | |
|---|---|---|---|---|---|---|
| | $\sigma_{D/N}$=0.5 | $\sigma_{D/N}$=1.0 | $\sigma_{D/N}$=2.0 | | $\sigma_{D/N}$=0.5 | $\sigma_{D/N}$=1.0 | $\sigma_{D/N}$=2.0 |
| $\sigma_N$=0.5 | 0.98% | 0.84% | 1.48% | $\sigma_N$=0.5 | 1.61% | 1.70% | 1.39% |
| $\sigma_N$=1.0 | 0.59% | **0.77%** | 0.91% | $\sigma_N$=1.0 | 1.09% | **0.96%** | 1.04% |
| $\sigma_N$=2.0 | 0.97% | 0.72% | 0.76% | $\sigma_N$=2.0 | 1.24% | 1.91% | 1.06% |

| **Setting (3)**, baseline 4.98% | | | | **Setting (4)**, baseline 10.54% | | |
|---|---|---|---|---|---|---|
| | $\sigma_{D/N}$=0.5 | $\sigma_{D/N}$=1.0 | $\sigma_{D/N}$=2.0 | | $\sigma_{D/N}$=0.5 | $\sigma_{D/N}$=1.0 | $\sigma_{D/N}$=2.0 |
| $\sigma_N$=0.5 | 2.09% | 0.99% | 1.60% | $\sigma_N$=0.5 | 5.16% | 4.05% | 5.04% |
| $\sigma_N$=1.0 | 2.53% | **3.03%** | 2.25% | $\sigma_N$=1.0 | 6.06% | **3.61%** | 5.41% |
| $\sigma_N$=2.0 | 2.29% | 2.72% | 2.58% | $\sigma_N$=2.0 | 6.54% | 3.80% | 3.72% |

As shown in Table 3, the default $\sigma = (1, 1)$ is within $\sim$1 pp of the best cell in every setting and is the best in (2). All nine cells beat the corresponding uniform-weight baseline in settings (2)–(4), so the surface is wide and forgiving in the data-constrained regime that the paper targets.

