# OpenReview forum: "Scaling Laws: A Model-Based Optimization Perspective"
_TMLR — Under review for TMLR_

### Review · Reviewer_Jjud · 2026-06-01

**Summary Of Contributions:**

**Summary**

This position paper interprets neural scaling laws from a Model-Based Optimization (MBO) perspective. The authors argue that the standard scaling-law pipeline (collecting small-scale runs, fitting a surrogate model, and optimizing configurations under a compute budget) can be viewed as an MBO problem. The paper further organizes the design space into model, data, and hyperparameter subspaces, and provides a proof-of-concept experiment using autofocus to improve scaling-law extrapolation.

**Strengths**

The central perspective is reasonable and useful. Framing scaling laws as MBO provides a clean way to connect existing scaling-law practice with broader tools such as uncertainty quantification, active experiment selection, and multi-objective optimization. The discussion of the design space is also helpful, especially the separation of model, data, and hyperparameter variables and the bi-level view of hyperparameter tuning. The autofocus experiment, while limited, gives a concrete example of how an MBO-inspired method can improve extrapolation.

**Weaknesses**

1. The limitations of the proposed MBO framing are not discussed sufficiently. The paper currently emphasizes the benefits of the MBO perspective, but a stronger position paper should also explain where MBO may break down.

2. The proof-of-concept experiment is helpful, but the implementation details are not fully specified. The authors should clarify the baseline fitting procedure, optimizer, initialization, stopping criteria, and autofocus hyperparameters. Additionally, reporting std. across runs would also improve reproducibility.

3. The paper could better explain how the distinctive properties of neural scaling are incorporated into the MBO framework. Although strong power-law priors are identified as a key difference from standard MBO, the proof-of-concept mainly applies autofocus reweighting on top of an existing parametric scaling-law surrogate. It would be useful to discuss how such priors should influence the MBO pipeline.

**Audience:**

Yes

**Audience Explanation:**

Neural scaling laws are increasingly important for foundation-model training, where compute allocation, model/data scaling, and extrapolation accuracy are practically significant problems. The paper provides a useful conceptual framing by connecting scaling-law practice to Model-Based Optimization, which may be relevant to researchers working on scaling laws, AutoML, efficient training, hyperparameter optimization, and data mixture optimization.

**Claims And Evidence:**

Yes

**Claims Explanation:**

The empirical evidence is limited but appropriate for the paper’s position-paper framing. The autofocus experiment provides a concrete proof of concept showing that an MBO-inspired method can improve scaling-law extrapolation in data-constrained settings. This supports the claim that MBO tools can be useful for neural scaling.

**Requested Changes:**

1. Add a short limitations or boundary-conditions section. The paper should discuss when the MBO framing is expected to be most useful, and when it may require additional assumptions or experiments (corresponding to the weakness 1).

2. Provide more experimental details for the autofocus proof of concept (corresponding to the weakness 2)

3. Strengthen the discussion of how neural-scaling-specific properties are incorporated into the MBO framework (corresponding to the weakness 3).

---

> ### Author Response · Authors · 2026-06-21
> **Rebuttal 1/1**
>
> Dear Reviewer,
>
> We sincerely appreciate your insightful comments, which have significantly strengthened our work. The manuscript has been revised accordingly, and all modifications are marked in red for your convenience.
>
> > The limitations of the proposed MBO framing are not discussed sufficiently. The paper currently emphasizes the benefits of the MBO perspective, but a stronger position paper should also explain where MBO may break down.
> >Add a short limitations or boundary-conditions section. The paper should discuss when the MBO framing is expected to be most useful, and when it may require additional assumptions or experiments (corresponding to the weakness 1).
>
> A standard MBO workflow assumes a single unified surrogate that maps designs to objective values; in our setting, the natural analogue would be a unified scaling-law formula $\mathcal{L}(\boldsymbol{x}_m, \boldsymbol{x}_d, \boldsymbol{x}_h)$ that holds across all architectures and modalities. The principal limitation of the MBO framing is that **we currently do not have a unified functional form**: different architectural and modality choices induce fundamentally different surrogate forms — the Chinchilla form $\mathcal{L} = E + A/N^\alpha + B/D^\beta$ for dense transformers, the bilinear MoE law of Clark et al. for routed models, and the multimodal laws of Shukor et al. all describe different parametric families that cannot be easily folded into a unified expression.
>
> We have stated this limitation explicitly as a short paragraph at the end of §2.3.
>
> > The proof-of-concept experiment is helpful, but the implementation details are not fully specified. The authors should clarify the baseline fitting procedure, optimizer, initialization, stopping criteria, and autofocus hyperparameters. Additionally, reporting std. across runs would also improve reproducibility.
> > Provide more experimental details for the autofocus proof of concept (corresponding to the weakness 2)
>
> We have added a short **Appendix A** stating the remaining implementation details:
>
> - **Log-space surrogate parameterisation.** We fit $\log\hat{\mathcal L} = \mathrm{logsumexp}(\log E,\;\log A - \alpha\log N,\;\log B - \beta\log D)$ rather than $\hat{\mathcal L}$ directly; optimisation variables are $(\log E, \log A, \alpha, \log B, \beta)$.
> - **Optimiser.** L-BFGS-B with `ftol=1e-15`, `gtol=1e-12`, `maxiter=5000`. Bounds: $\log E\in[\log 0.01,\log 10]$; $\log A,\log B\in[\log 0.01,\log 10^6]$; $\alpha,\beta\in[0.01,2]$.
> - **15-restart ensemble and reproducibility.** The 15 random restarts already mentioned in §4.3 draw from $E\sim\mathcal U(0.3,3)$, $A,B\sim\mathcal U(1,100)$, $\alpha,\beta\sim\mathcal U(0.1,1)$ via a fixed `np.random.RandomState(42)`. To verify stability, we re-ran the entire 15-restart ensemble pipeline under 5 independent master seeds; the mean absolute percentage error fluctuates by less than 0.5pp across runs in every setting, so the reported numbers are reproducible up to negligible noise.
>
> > The paper could better explain how the distinctive properties of neural scaling are incorporated into the MBO framework. Although strong power-law priors are identified as a key difference from standard MBO, the proof-of-concept mainly applies autofocus reweighting on top of an existing parametric scaling-law surrogate. It would be useful to discuss how such priors should influence the MBO pipeline.
> > Strengthen the discussion of how neural-scaling-specific properties are incorporated into the MBO framework (corresponding to the weakness 3).
>
> We have added a new paragraph in §4.5 ("Discussion: Validating the MBO Paradigm") tying the power-law prior to the autofocus PoC:
>
> "This perspective shift becomes operative precisely because of neural scaling's strong functional prior. Standard MBO surrogates are universal approximators (e.g., MLPs); locally up-weighting points around a target with such a model simply specializes it to the local region and loses extrapolation. By contrast, the rigid 5-parameter power-law surrogate (Chinchilla form) constrains the fit to a global functional form, so autofocus's local re-weighting only re-positions the few free parameters to better capture the local regime — exactly the property that makes the MBO toolkit applicable here."

---

### Review · Reviewer_ouKc · 2026-06-03

**Summary Of Contributions:**

### Summary

This paper presents a new perspective that frames neural scaling laws as a special instance of Model-Based Optimization (MBO). The authors map the standard scaling-law workflow to three stages of MBO: defining the design space, fitting a surrogate model, and performing guided optimization under compute constraints. The paper further organizes the scaling-law design space into model, data, and hyperparameter subspaces, and formulates the interaction between model/data choices and hyperparameter tuning as a bi-level optimization problem. To illustrate the usefulness of the MBO perspective, the paper applies an autofocus-style importance weighting method to scaling-law surrogate fitting and shows improved extrapolation accuracy in data-constrained settings.

### Strengths

1. The paper provides a coherent mapping between neural scaling laws and MBO. This perspective helps clarify that scaling laws are not merely empirical curve fits, but surrogate models used for downstream optimization under strict compute budgets.

2. The paper gives a broad and well-structured taxonomy of the relevant design variables, including model architecture and scale, data composition and scale, and training hyperparameters. The bi-level optimization formulation is also a reasonable way to express the fact that scaling laws usually assume well-tuned hyperparameters for each model/data configuration.

3. The autofocus experiment provides a useful example of how an existing MBO technique can be transferred to scaling-law extrapolation.

### Weaknesses

1.    While the MBO framing is intuitive and potentially useful, the paper does not propose a unified MBO-based scaling law that jointly models the model, data, and hyperparameter subspaces. Many variables are discussed in the taxonomy, but they are not integrated into a single operational surrogate model or optimization procedure. As a result, the paper sometimes reads more like a relabeling and organization of existing scaling-law practices than a fully developed new methodology.

2. The autofocus experiment is helpful as a proof-of-concept, but it is conducted on a constrained public dataset and targets relatively small model scales compared with modern frontier scaling-law applications.

3.  The effectiveness of the proposed autofocus method depends on how the target distribution is defined, including the choice of feature space, normalization, kernel form, and bandwidth. A more systematic sensitivity analysis or theoretical justification would strengthen the proposed method.

**Audience:**

Yes

**Audience Explanation:**

Scaling laws have achieved great attention over past years, both empirically and theoretically. A new model-based optimization perspective would be of great interest to researchers in optimization and LLMs within the TMLR community.

**Claims And Evidence:**

Yes

**Claims Explanation:**

The main claims of the submission are supported by reasonably clear evidence.

1.
   The paper provides a clear conceptual mapping between the standard neural scaling-law pipeline and the three stages of Model-Based Optimization: design space definition, surrogate modeling, and guided optimization. In particular, the paper explains how pre-training configurations can be viewed as design variables, validation loss or benchmark performance as labels, fitted scaling laws as surrogate models, and compute-constrained configuration search as guided optimization. This theoretical correspondence is presented clearly and is largely convincing.

2.
   The paper further supports its argument with an illustrative experiment using autofocus, an existing MBO technique, for scaling-law extrapolation. The experiment shows that reweighting training points according to their relevance to the target configuration can improve extrapolation accuracy, especially in data-constrained regimes.

**Requested Changes:**

### Requested Changes

1. The autofocus experiment provides useful evidence that MBO techniques can improve scaling-law extrapolation. At the same time, the current experiment is relatively limited in scale and scope. I suggest adding a more explicit discussion of this limitation, including whether the authors expect similar behavior for larger models, larger token budgets, or more realistic pretraining settings.

2. The autofocus method depends on the choice of target distribution, including the feature representation, normalization, kernel form, and bandwidth. The current choices are reasonable and intuitive, but readers would benefit from more explanation of why these choices were made and how they should be adapted in other scaling-law settings. Even a qualitative discussion  would make the method easier to interpret and apply.

---

> ### Author Response · Authors · 2026-06-21
> **Rebuttal 1/2**
>
> Dear Reviewer,
>
> Thank you for your thoughtful feedback, which has greatly helped us improve our paper. We have carefully revised the manuscript in response to your suggestions, with all changes highlighted in red for ease of review.
>
> ## WEAKNESSES
>
> > While the MBO framing is intuitive and potentially useful, the paper does not propose a unified MBO-based scaling law that jointly models the model, data, and hyperparameter subspaces. Many variables are discussed in the taxonomy, but they are not integrated into a single operational surrogate model or optimization procedure. As a result, the paper sometimes reads more like a relabeling and organization of existing scaling-law practices than a fully developed new methodology.
>
>
> We propose precisely such a unified law at the *structural* level — Eq. (2) in §3, the bi-level form. This is the general MBO scaling law: it specifies how the three subspaces interact, without committing to a single closed-form surrogate.
>
> A single *specific* closed-form surrogate jointly over $(\boldsymbol{x}_m, \boldsymbol{x}_d, \boldsymbol{x}_h)$ is currently out of reach, for three reasons:
>
> 1. **The lower-level map $\boldsymbol{x}_h^*(\boldsymbol{x}_m, \boldsymbol{x}_d)$ is implicit.** Optimal hyperparameters as a function of architecture and data are governed by training dynamics; no explicit functional form is known across the full $(\boldsymbol{x}_m, \boldsymbol{x}_d)$ space.
> 2. **The upper-level surrogate is not architecture-agnostic.** The Chinchilla form $L = E + A/N^\alpha + B/D^\beta$ (dense transformers), the bilinear MoE law of Clark et al. (routed models) and the multimodal laws of Shukor et al. all have fundamentally different functional forms; a single form unifying them does not currently exist.
> 3. **Even in the simplest text-only setting, the coefficient-level dependence is open.** Within the Chinchilla form, $(E, A, \alpha, B, \beta)$ are known to depend on the design variables, but the specific functional form of that dependence remains an empirical open question.
>
> **Position-paper scope.** Our contribution is to show that the existing scaling-law zoo — Chinchilla, bilinear MoE, data-mixture, precision-aware — composes naturally under Eq. (2)'s bi-level structure, and that the MBO framing makes this composition explicit rather than implicit. Identifying the unified *specific* surrogate would require a coordinated empirical campaign spanning architecture, modality, mixture, and precision under one compute-equivalent grid — substantial compute and engineering beyond the scope of a position paper.
>
> > The autofocus experiment is helpful as a proof-of-concept, but it is conducted on a constrained public dataset and targets relatively small model scales compared with modern frontier scaling-law applications.
> > The autofocus experiment provides useful evidence that MBO techniques can improve scaling-law extrapolation. At the same time, the current experiment is relatively limited in scale and scope. I suggest adding a more explicit discussion of this limitation, including whether the authors expect similar behavior for larger models, larger token budgets, or more realistic pretraining settings.
>
> We address them jointly with a new limitations paragraph at the end of §4:
>
> **Limitations and Expected Behavior at Scale.** Scaling laws are approximations of the true loss landscape; autofocus's contribution is to upweight training points near the prediction target when fitting the surrogate. This principle does not depend on absolute scale, so we expect the same qualitative behavior at (i) larger model sizes, (ii) larger token budgets. Empirically validating this carry-over at frontier scales (≥10B parameters, trillion-token budgets) would require pre-training data we do not have access to, and no public dataset of comparable diversity exists at those scales. We therefore deliberately limit the empirical commitment of this position paper to a verifiable open dataset (Porian et al. 2024) and flag frontier-scale validation as the natural empirical follow-up.

---

> > ### Author Response · Authors · 2026-06-21
> > **Rebuttal 2/2**
> >
> > > The effectiveness of the proposed autofocus method depends on how the target distribution is defined, including the choice of feature space, normalization, kernel form, and bandwidth. A more systematic sensitivity analysis or theoretical justification would strengthen the proposed method.
> > > The autofocus method depends on the choice of target distribution, including the feature representation, normalization, kernel form, and bandwidth. The current choices are reasonable and intuitive, but readers would benefit from more explanation of why these choices were made and how they should be adapted in other scaling-law settings. Even a qualitative discussion would make the method easier to interpret and apply.
> >
> > The reviewer names some design choices: **feature space, normalization, kernel form, and bandwidth**. The feature-space comparison is already in the submission (§4.4 obs. (3): $(\log N, \log D/N)$ vs $(\log N, \log D)$). For the other three we ran new ablations on the same dataset and protocol as §4.3; all numbers below are mean |% error| on the held-out 14 test points. The three tables have appeared as a new Appendix B in the revision.
> >
> > **(a) Normalization — min-max (paper) vs z-score vs none.** Same Gaussian kernel, $\sigma=(1, 1)$, $\delta=10^{-3}$, features $(\log N, \log D/N)$.
> >
> > | Setting | Baseline | min-max (paper) | z-score | none |
> > |---------|---------:|----------------:|--------:|-----:|
> > | (1) |  0.50% | **0.77%** | 1.33% | 0.88% |
> > | (2) |  2.98% | **0.96%** | 1.59% | 1.60% |
> > | (3) |  4.98% | 3.03% | 2.56% | **2.14%** |
> > | (4) | 10.54% | **3.61%** | 7.11% | 6.25% |
> >
> > Min-max wins in three of four settings and is by far the best in the hardest setting (4), where the z-score and no-normalization variants degrade sharply (7.11%/6.25%). The reason min-max is preferable is that it bounds both features into the same $[0, 1]$ range, so a single default $\sigma$ has consistent meaning across settings; z-score's std-based scale shifts as the training pool shrinks, and the raw-log variant gives the two features very different effective ranges.
> >
> > **(b) Kernel form — Gaussian (paper) vs Epanechnikov $K(u) = \max\\big(\tfrac{3}{4}(1-u^2)\,0\big)$.** Same $\sigma=1$ in normalized $[0,1]$ feature space.
> >
> > | Setting | Baseline | Gaussian (paper) | Epanechnikov |
> > |---------|---------:|-----------------:|-------------:|
> > | (1) |  0.50% | **0.77%** | 1.01% |
> > | (2) |  2.98% | **0.96%** | 1.06% |
> > | (3) |  4.98% | 3.03% | **2.68%** |
> > | (4) | 10.54% | **3.61%** | 4.34% |
> >
> > Both local-kernel variants beat the uniform baseline in the data-constrained settings (2)–(4). Gaussian wins in (1), (2), and (4); Epanechnikov wins in (3). The takeaway is that *any* smooth local re-weighting captures the autofocus benefit — the choice of kernel matters less than the *fact* of localization.
> >
> > **(c) Bandwidth $\sigma$ — 3×3 grid over $\sigma_N, \sigma_{D/N} \in \{0.5, 1.0, 2.0\}$**, paper's defaults bolded.
> >
> > |        Setting (1) | $\sigma_{D/N}=0.5$ | $1.0$ | $2.0$ |
> > |:------------------:|------:|------:|------:|
> > | $\sigma_N = 0.5$   | 0.98% | 0.84% | 1.48% |
> > | $\sigma_N = 1.0$   | 0.59% | **0.77%** | 0.91% |
> > | $\sigma_N = 2.0$   | 0.97% | 0.72% | 0.76% |
> >
> > |        Setting (2) | $\sigma_{D/N}=0.5$ | $1.0$ | $2.0$ |
> > |:------------------:|------:|------:|------:|
> > | $\sigma_N = 0.5$   | 1.61% | 1.70% | 1.39% |
> > | $\sigma_N = 1.0$   | 1.09% | **0.96%** | 1.04% |
> > | $\sigma_N = 2.0$   | 1.24% | 1.91% | 1.06% |
> >
> > |        Setting (3) | $\sigma_{D/N}=0.5$ | $1.0$ | $2.0$ |
> > |:------------------:|------:|------:|------:|
> > | $\sigma_N = 0.5$   | 2.09% | 0.99% | 1.60% |
> > | $\sigma_N = 1.0$   | 2.53% | **3.03%** | 2.25% |
> > | $\sigma_N = 2.0$   | 2.29% | 2.72% | 2.58% |
> >
> > |        Setting (4) | $\sigma_{D/N}=0.5$ | $1.0$ | $2.0$ |
> > |:------------------:|------:|------:|------:|
> > | $\sigma_N = 0.5$   | 5.16% | 4.05% | 5.04% |
> > | $\sigma_N = 1.0$   | 6.06% | **3.61%** | 5.41% |
> > | $\sigma_N = 2.0$   | 6.54% | 3.80% | 3.72% |
> >
> > The default $\sigma=(1,1)$ is within ~1 pp of the best cell in every setting and is the best in (2). All 9 cells beat the uniform baseline in settings (2)–(4), so the surface is wide and forgiving in the data-constrained regime that the paper targets.